# Cortical Reorganization of Early Somatosensory Processing in Hemiparetic Stroke

**DOI:** 10.3390/jcm11216449

**Published:** 2022-10-31

**Authors:** Jordan N. Williamson, William A. Sikora, Shirley A. James, Nishaal J. Parmar, Louis V. Lepak, Carolyn F. Cheema, Hazem H. Refai, Dee H. Wu, Evgeny V. Sidorov, Julius P. A. Dewald, Yuan Yang

**Affiliations:** 1Neural Control and Rehabilitation Laboratory, Stephenson School of Biomedical Engineering, University of Oklahoma, Norman, OK 73019, USA; 2Department of Biostatistics and Epidemiology, College of Public Health, University of Oklahoma Health Sciences Center, Oklahoma City, OK 73104, USA; 3Department of Electrical and Computer Engineering, University of Oklahoma, Tulsa, OK 74135, USA; 4Department of Rehabilitation Sciences, University of Oklahoma Health Sciences Center, Tulsa, OK 74135, USA; 5Department of Radiological Sciences, University of Oklahoma Health Sciences Center, Oklahoma City, OK 73104, USA; 6Department of Neurology, University of Oklahoma Health Sciences Center, Oklahoma City, OK 73104, USA; 7Department of Physical Therapy and Human Movement Sciences, Feinberg School of Medicine, Northwestern University, Chicago, IL 60607, USA; 8Department of Biomedical Engineering, McCormick School of Engineering, Northwestern University, Evanston, IL 60208, USA

**Keywords:** hemiparetic stroke, cortical reorganization, somatosensory evoked potentials, EEG, sensorimotor system

## Abstract

The cortical motor system can be reorganized following a stroke, with increased recruitment of the contralesional hemisphere. However, it is unknown whether a similar hemispheric shift occurs in the somatosensory system to adapt to this motor change, and whether this is related to movement impairments. This proof-of-concept study assessed somatosensory evoked potentials (SEPs), P50 and N100, in hemiparetic stroke participants and age-matched controls using high-density electroencephalograph (EEG) recordings during tactile finger stimulation. The laterality index was calculated to determine the hemispheric dominance of the SEP and re-confirmed with source localization. The study found that latencies of P50 and N100 were significantly delayed in stroke brains when stimulating the paretic hand. The amplitude of P50 in the contralateral (to stimulated hand) hemisphere was negatively correlated with the Fügl–Meyer upper extremity motor score in stroke. Bilateral cortical responses were detected in stroke, while only contralateral cortical responses were shown in controls, resulting in a significant difference in the laterality index. These results suggested that somatosensory reorganization after stroke involves increased recruitment of ipsilateral cortical regions, especially for the N100 SEP component. This reorganization delays the latency of somatosensory processing after a stroke. This research provided new insights related to the somatosensory reorganization after stroke, which could enrich future hypothesis-driven therapeutic rehabilitation strategies from a sensory or sensory-motor perspective.

## 1. Introduction

Stroke is the leading cause of serious, long-term disability in adult individuals. Approximately 80% of stroke survivors report movement impairment on the side of the body contralateral to the lesioned hemisphere [1]. Despite the development of many interventions for motor recovery after a stroke, rehabilitation treatments, especially in individuals with more severe impairments, are only partially effective [2,3,4]. The potential for more effective and targeted treatment relies on a better understanding of neural circuitry changes in the brain after a stroke and during recovery [4,5]. 

Previous neuroimaging studies after hemiparetic stroke have shown that movement of the paretic arm is often associated with increased activity in the contralesional (ipsilateral to the paretic side) motor cortices [6,7,8]. The increased activity ipsilateral to brain lesion motor is likely related to a greater reliance on ipsilateral cortico–bulbospinal pathways following stroke-induced damage to contralateral motor pathways at the lesioned hemisphere [7]. Previous studies found that increased reliance on contralesional descending cortico–reticulospinal pathways [7,9,10,11,12,13] likely accounts for post-stroke movement impairment such as abnormal flexion synergy [14] and spasticity [13,15,16]. 

The control of movement requires somatosensory feedback. However, how the somatosensory system adapts to the change in the use of motor pathways and the role of adaptive sensory feedback to the abnormal movement control of the paretic arm remains largely unknown. The ascending sensory pathways that convey somatosensation from the paretic arm project contralaterally to the primary sensory cortex in the lesioned hemisphere. It is unknown, however, whether a similar hemispheric shift in cortical somatosensory processing after a stroke occurs may be related to the maladaptive use of contralesional cortico–reticulospinal pathways and motor impairment [11]. The answer to this question is important since it may permit a potential assessment of motor deficits from a sensory perspective, which could be clinically significant in more severely impaired individuals who can barely perform any functional movement tasks, as well as in individuals in the acute/subacute phases of recovery from a stroke whose movement ability is still limited or absent. This also prevents “over-exerting” a more impaired individual or an acute individual while performing motor assessments or strenuous non-targeted rehabilitative interventions, thus encouraging the maladaptive use of reticulospinal pathways resulting in the emergence and expression of the flexion synergy and spasticity after a stroke [17].

To explore this question, this proof-of-concept study assessed the cortical somatosensory processing in chronic stroke patients and compared it with that in age-matched control subjects. The electroencephalogram (EEG) was recorded when the participants are receiving electrical tactical index finger stimulation to investigate cortical somatosensory processing based on somatosensory evoked potentials (SEP) and source localization. Electrical stimulation of the index finger was selected because we aimed to target exclusively Aβ sensory fibers. Aβ fibers provide pure tactile sensory information, compared to the commonly stimulated, more proximal portion of the median nerve at the palm or forearm that provides both sensory (tactile and muscle afferents) and motor activity to the forearm, wrist, and hand muscles [18,19]. Cutaneous Aβ fibers, even though thicker than Aδ and C fibers, are thinner than group I and II muscle afferents and stimulated more distally at the index finger, thus resulting in a longer time delay to the primary motor cortex of greater than 20 ms [20]. Therefore, based on the literature, components P50 and N100 of the SEP were selected as time points of analysis since they are the earliest SEP components where little integration from other cortical areas took place, yet long enough to get a SEP response not contaminated by the stimulation artifact [21,22,23,24].

## 2. Materials and Methods

Nine individuals’ post-stroke (three females) and eight age-matched healthy controls (four females) participated in this study. The study is approved by the internal review board (IRB) of the University of Oklahoma Health Sciences Center (IRB # 12550). The demographics of stroke participants are provided in Table 1, including participants’ Fügl–Meyer upper extremity scores (FM-UE) [25].

Subjects’ index fingers were stimulated using Digitimer DS7A Constant Current Stimulator (Digitimer Ltd., Welwyn Garden City, UK). The electrodes were placed with the positive and ground termini on the distal and intermediate phalanges on the index finger, respectively, as displayed in Figure 1. Stimulation was applied first to the paretic and then non-paretic hand in the stroke group to allow for within-subject comparisons. Stimulation was applied to the dominant hand of control participants. The stimulus was delivered in the form of a DC square wave with a duration of 200 µs and current normalized to twice the sensation threshold for each participant. Stroke participants had a significantly higher sensation threshold than healthy subjects in their paretic hands (two-sample *t*-test *p* = 0.025), resulting in higher actual stimulation intensity. There was no significant difference in sensation threshold or actual stimulation intensity between the tested hand in healthy controls and the non-paretic hand in stroke. Each trial was one minute in duration, consisting of 120 individual stimuli delivered at 2 Hz, and 5 trials were conducted for each participant. 

Brain response data was collected using the BrainVision Recorder EEG System (Brain Vision LLC, Morrisville, NC, USA). An EasyCap electrode cap (EASYCAP GmbH, Woerthsee-Etterschlag, Germany) of the correct size for each participant was fitted with 64 electrodes in the 10–20 system. A sampling rate of at least 1000 Hz was used to collect all data, and a software notch filter was enabled at 60 Hz to mitigate interference by the electrical grid. 

Data analysis was conducted in EEGLAB [26] for MATLAB R2020a (MathWorks, Natick, MA, USA). First, all trials were appended to each other. The data were visually inspected, and noisy or otherwise unsuitable channels were removed. After bandpass filtering between 1 and 45 Hz, each dataset was re-referenced to the global average reference of all remaining channels and epoched with a window of −80 to 300 ms surrounding each stimulus. Epoch baselines were calculated from −80 to 0 ms before the stimulus and removed. A notable artifact of stimulation was observed in each participant along a window from 0 to 2.5 ± 0.3 ms after stimulus. This unique interval was identified for each participant, both stroke and controls, and replaced with a cubic interpolation of the data for 50 ms on either side of the window. 

Epochs were then visually inspected and rejected based on the presence of blinking and movement artifacts, leaving 300 epochs on average per participant. The epochs were then averaged in each participant to extract the somatosensory evoked potentials (SEP). The latency and amplitude of early SEP components, P50 and N100, were measured at both contralateral and ipsilateral hemispheres around the sensorimotor areas, i.e., C3/4, C5/6, C1/2, CP3/4, CP5/6, CP1/2. For each participant, the latency of each component was taken at the electrode where the amplitude was maximal over each hemisphere, and the amplitude was measured at each electrode over the same hemisphere at that latency. ERP voltage maps were calculated and drawn at the mean latency of each component. The standardized low-resolution electromagnetic tomographic analysis sLORETA (v20200701) was used to localize the ERP source activity on the cortex [27,28].

The laterality index was computed to investigate the hemisphere dominance of cortical response in the time window of the P50-N100 [12,29]. The LI is defined as the signal power difference between contralateral and ipsilateral hemispheres in the sensorimotor areas (including C1/2, C3/4, C5/6, CP1/2, CP3/4, CP5/6 in 10/20 EEG recording system) and then normalized by their sum, as shown in the equation below. A higher LI indicates a stronger contralateral dominance (healthy normal) while a reduced LI indicates either more bilateral activities or an ipsilateral dominance (if LI < 0) that is likely due to functional reorganization in the brain.
(1)$$\mathrm{LI}=\frac{\mathrm{Contralateral}-\mathrm{Ipsilateral}}{\mathrm{Contralateral}\;+\;\mathrm{Ipsilateral}}$$

Statistical analyses were performed using commercial software Statistical Analysis Systems (9.4, SAS, Carey, NC, USA). First, an independent *t*-test was performed to ensure the stroke participants and controls had a similar age range (50–80 years, two-sample *t*-test *p* = 0.23). Analysis of variance (ANOVA) was performed to check the statistical significance of the results using stimulation category (stroke paretic vs. stroke non-paretic vs. controls) for ERP latencies, mean amplitudes, and mean laterality index. Then summary statistics were computed including means, 95% CI, medians, and standard deviations (Table 2). We checked the outcome variables to assure they were normally distributed; and found no evidence to the contrary. We then analyzed the data using correlated data analysis with generalized estimating equations (GEE) (PROC GENMOD) to produce correlated linear models for each outcome variable. We utilized GEE analysis because it offers robust beta estimates despite variance structure specification. Because two of our comparisons were correlated (stroke-involved and stroke uninvolved arms), and one was not (control), this methodology allowed for comparison of the correlated data. We performed separate GEE analyses using the stimulation category (stroke paretic vs. stroke non-paretic vs. controls) for ERP latencies, mean amplitudes, and mean laterality index. We then completed Pearson correlation analyses between ERP latencies and amplitude and motor impairment levels. 

## 3. Results

Visualization of the contralateral and ipsilateral (to stimulated hand) SEP responses to finger stimulation are shown in Figure 2 and Figure 3. The contralateral SEPs (P50 and N100) were shown in both stroke and control participants, while the ipsilateral SEPs were mainly shown in stroke participants when their paretic hand was stimulated. 

The descriptive statistics of the latency, amplitude, and laterality index are displayed in Table 2. In the contralateral (to stimulated hand) hemisphere, the ANOVA results showed that the latencies of P50 (F (2,22) = 12.71, *p* < 0.0002) and N100 (F (2,22) = 10.06, *p* < 0.0008) were significantly different between groups. Individual GEE analysis showed that the latency of P50 was significantly delayed in both the paretic hand (z = 4.76, *p* = <0.0001) and the non-paretic hand (P50 z = 3.33, *p* = 0.0009) compared to the controls. Additionally, at timepoint N100, the stroke paretic hand (z = 4.16, *p* = <0.0001) was significantly delayed compared to controls. For stroke participants, within-subject comparisons show that the latencies of P50 (paretic vs. nonparetic: z = 2.82, *p* = 0.0047) and N100 (z = 3.44, *p* = 0.0006) were larger for stimulation at paretic hand than nonparetic (see Figure 4 and Figure 5).

The amplitude differences of P50 and N100 in the contralateral (to stimulated hand) hemisphere were not statistically significant between stroke and control groups. The mean values of amplitude are reported in Table 2. The Pearson correlation analysis showed that there was a significant negative linear relationship between the P50 amplitude of the contralateral (to stimulated hand) SEP responses and Fügl–Meyer upper extremity (FM-UE) score (R = −0.630, *p* = 0.047), as shown in Figure 6. No significant correlations were found between FM-UE and other SEP measures. 

The laterality index (Figure 7) was significantly lower when the stroke paretic hand was stimulated compared to the stroke non-paretic hand (z = −2.44, *p* = 0.033) and healthy control (*p* = 0.022), indicating more bilateral or ipsilateral cortical activities after a stroke. This was also evident in source localization results where only contralateral source activity was detected in healthy controls, which was in line with previous findings [23,30,31], while bilateral source activities were shown in individuals after a stroke (Figure 8, Figure 9 and Figure 10). 

## 4. Discussion

The laterality index and source localization results showed that bilateral cortical responses occurred in stroke participants when their paretic hand was stimulated, while controls had only unilateral cortical responses on the contralateral (to stimulated hand) hemisphere. The bilateral response in stroke participants was mostly seen at the timepoint of N100. These results suggest somatosensory reorganization occurs post-stroke. This reorganization is likely due to the increased recruitment of ipsilateral cortical regions during the processing of the somatosensory signals from the paretic hand. This is consistent with neuroimaging studies that have demonstrated increased ipsilateral cortical sensorimotor activity during movement [6,7,8,12], which may require the sensory signal to re-route to provide sensory feedback for ipsilateral motor control. 

The change in somatosensory neural circuitry might occur subcortically; however, there is no known ascending bilateral or ipsilateral pathway for carrying tactile signals from a distal periphery nerve to the somatosensory cortices. The ascending pathways in the dorsal column that convey tactile sensation from the paretic arm project contralaterally to the primary sensory cortex in the lesioned hemisphere. Therefore, a potential neural mechanism may be a crossover of signals in the corpus callosum. This would also explain the ipsilesional activity during the P50 and the more delayed contralesional N100 somaesthetic evoked potential following stimulation of the paretic index finger. The corpus callosum is the largest white matter pathway connecting the two cerebral hemispheres and has the role of mediating interhemispheric modulation between the primary motor cortex and sensory cortices to facilitate coordinated movements [32]. The assumption of its role in post-stroke somatosensory processing is based on existing knowledge that interhemispheric transfer of sensory information relies on the posterior half of the corpus callosum and the integrity of the sensory region is reduced in chronic stroke [32,33,34]. Additionally, other research has shown that bilateral activation of the primary somatosensory cortex occurs during mirror therapy post-stroke and the corpus callosum was found to be involved [35]. This interhemispheric transfer of sensory information can also explain the delayed latency of the N100 SEPs for stimulation of the stroke paretic hand as we reported in this study. The delayed latency at timepoint P50 is likely due to stroke-induced supraspinal damage of the dorsal columns (white matter stroke) since the source localization results show mostly activation over the lesioned hemisphere. 

Additionally, while not statistically significant, the reduced amplitudes are in line with prior studies on SEP’s post-stroke [36,37,38]. The negative linear relationship between P50 amplitude and Fügl–Meyer impairment shows that the degree of the motor impairments is related to the hemispheric shift in cortical responses of sensory information post-stroke. This is consistent with the literature as Keren, Ring [39] established a negative relationship between upper limb SEP with clinical performance. This information on the relationship between the change in SEP and motor impairment is clinically significant. While it is known that somatosensory deficits worsen the recovery of motor function and adding sensory stimulation in rehabilitation practices enhances motor recovery, sensory reorganization in an injured brain is not sufficiently considered in current clinical practices [40,41]. This information potentially helps predict the severity of motor impairment based on the degree of cortical activity to sensory stimulation after a stroke. If motor impairment could be gauged from a sensory perspective, this would help complete a more comprehensive assessment, especially in those individuals who can barely perform any upper limb movements. Additionally, directed rehabilitation interventions focusing on engaging the somatosensory tracts have the potential to enhance motor recovery for individuals in the acute/subacute phases of recovery whose movement ability is still limited or absent. This type of directed sensory rehabilitation is currently being explored, such as focal repetitive muscle vibration, which is a non-invasive post-stroke therapy to reduce muscle tone [42,43]. Another example is wearable focal stimulation devices, such as a vibrotactile glove (VTG), which provides vibratory input to the paretic limb of chronic stroke survivors and has been shown to promote neural plasticity and reduces spasticity [44,45]. Other studies have used robot-assisted somatosensory training and vibrotactile biofeedback devices [46,47]. 

In summary, this research provides new knowledge to further understand neural mechanisms underlying motor deficits induced by somatosensory reorganization after a hemiparetic stroke. This is significant because it will pave the way to providing a sensitive biomarker based on EEG to enrich future science-driven therapeutic rehabilitation strategies from a sensory or sensory-motor perspective, thus improving stroke recovery. 

Limitations and future work concern the lack of fine anatomical resolution in the EEG to determine the physical pathway of re-route somatosensory process in the brain, and the limited number of participants. While EEG boasts sufficient temporal resolution to elucidate the delay of action and reorganization of the somatosensory processing network in impaired stroke patients, it cannot be used to determine which pathway neural signals take from the contralateral to the ipsilateral cortex. Additionally, EEG provides very limited information on any changes in subcortical regions. Therefore, while we assume that the crossover in sensory signals occurs at the corpus callosum, the exact pathway remains hypothetical. Other modalities of neuroimaging, such as functional magnetic resonance imaging (fMRI) or diffusion tensor imagining might offer an improved ability to determine information flow in the brain in real time. In addition, future work could also involve simultaneous EEG-fMRI to provide a more precise interpretation of results. If this relationship is successfully established, it would further our understanding of neuroplasticity following unilateral brain injury. This would aid in improved rehabilitation strategies such as neurostimulation, which to this point has found very limited clinical adoption given its temporary effects. An additional aspect of this study that could be improved is the small sample size. Therefore, future work will focus on increasing the number of study participants. 

## Figures and Tables

**Figure 1 jcm-11-06449-f001:**
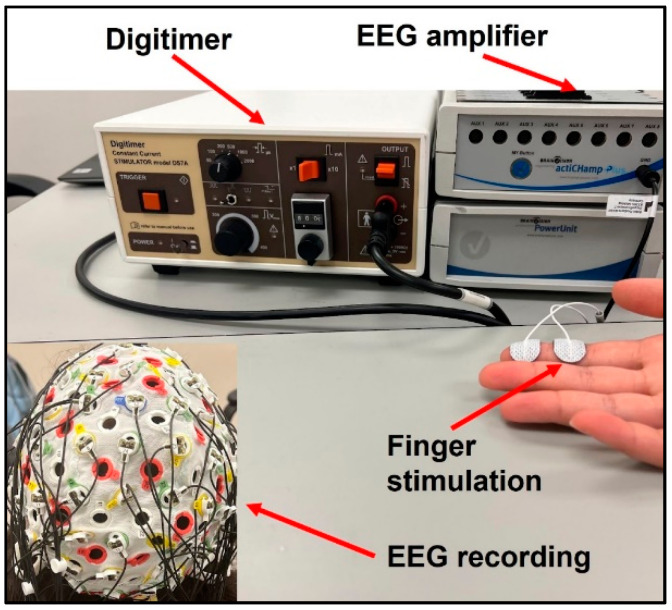
Experimental Setup.

**Figure 2 jcm-11-06449-f002:**
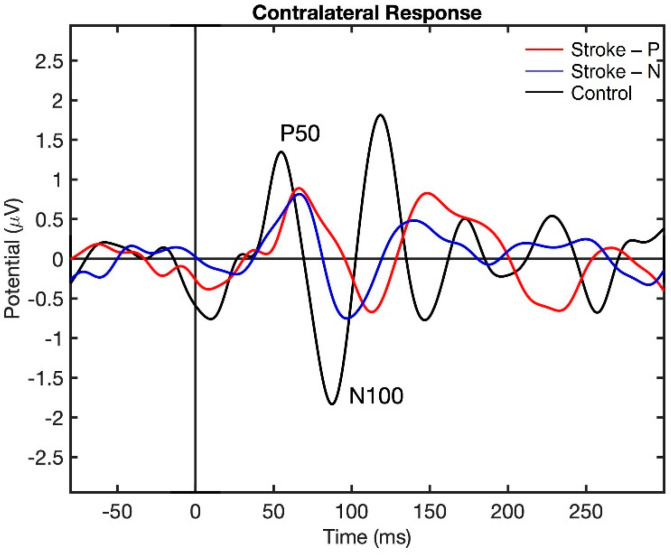
Contralateral Somatosensory Evoked Potential (SEP) response to finger stimulation. Stroke-P (red): paretic hand was stimulated. Stroke-N (blue): non-paretic hand was simulated. Control (black): dominant hand was stimulated.

**Figure 3 jcm-11-06449-f003:**
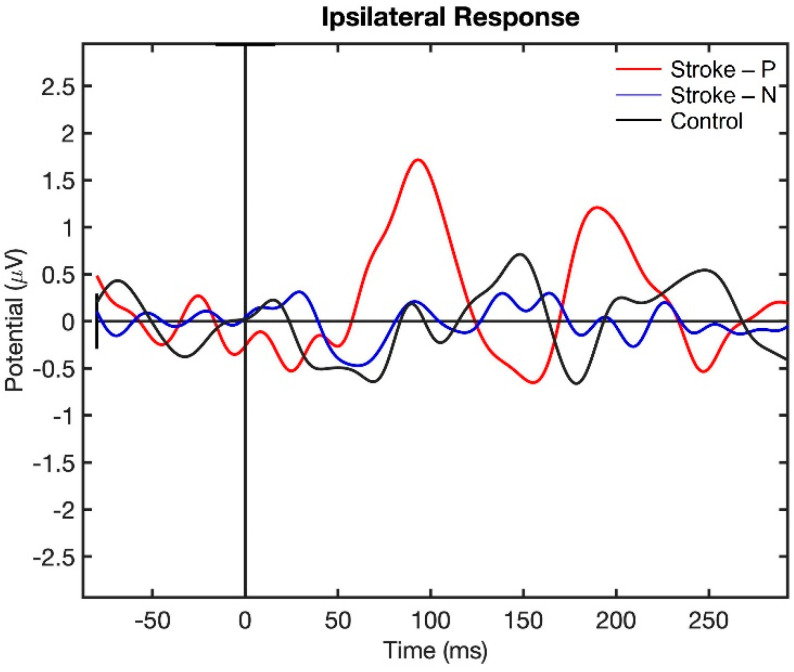
Ipsilateral Somatosensory Evoked Potential (SEP) response to finger stimulation. Stroke-P (red): paretic hand was stimulated. Stroke-N (blue): non-paretic hand was simulated. Control (Black): dominant hand was stimulated.

**Figure 4 jcm-11-06449-f004:**
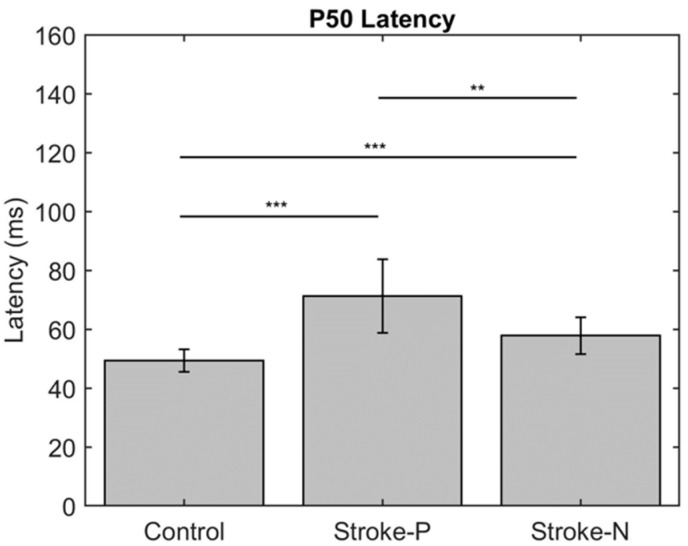
Latency of contralateral (to stimulated hand) Somatosensory Evoked Potential (SEP) component P50. Stars indicate statistically significant differences between groups (control, stroke paretic hand (Stroke-P) and stroke non-paretic hand (Stroke-N)) ** < 0.01 *** < 0.001.

**Figure 5 jcm-11-06449-f005:**
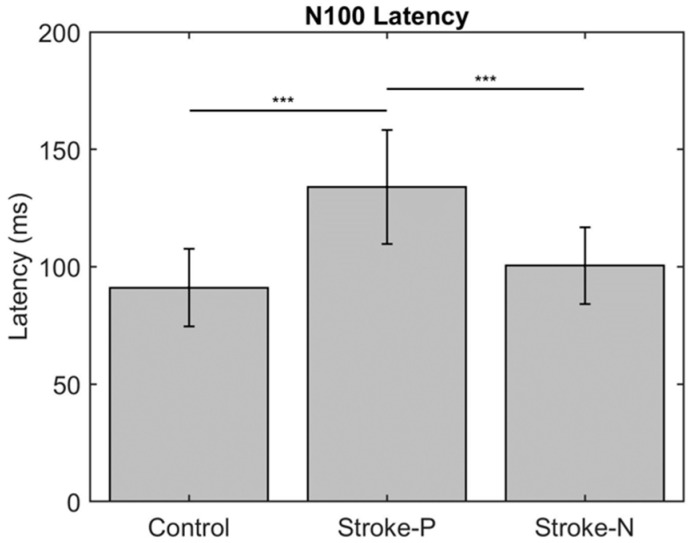
Latency of contralateral (to stimulated hand) Somatosensory Evoked Potential (SEP) component P50. Stars indicate a statistically significant difference between groups (control, stroke paretic hand (Stroke-P) and stroke non-paretic hand (Stroke-N)): *** < 0.001.

**Figure 6 jcm-11-06449-f006:**
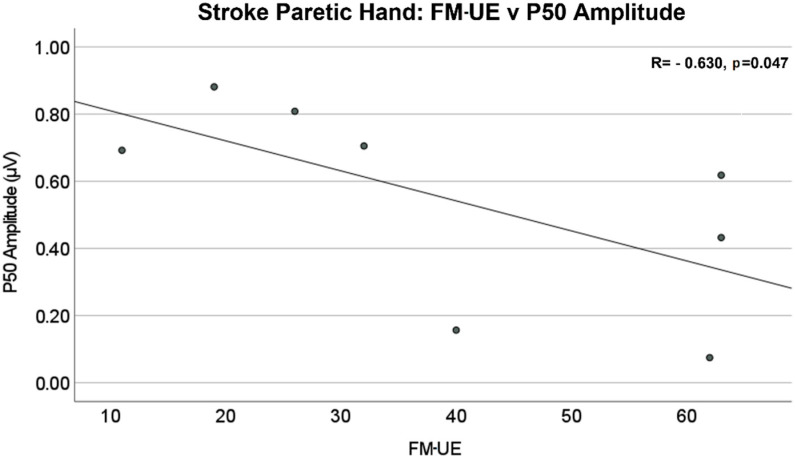
Stroke paretic hand: Fügl–Meyer upper extremity (FM-UE) score vs. P50 amplitude in the contralateral (to stimulated hand) hemisphere. There is a significant negative linear relationship between P50 amplitude and FM-UE Score (R = −0.630, *p* = 0.047).

**Figure 7 jcm-11-06449-f007:**
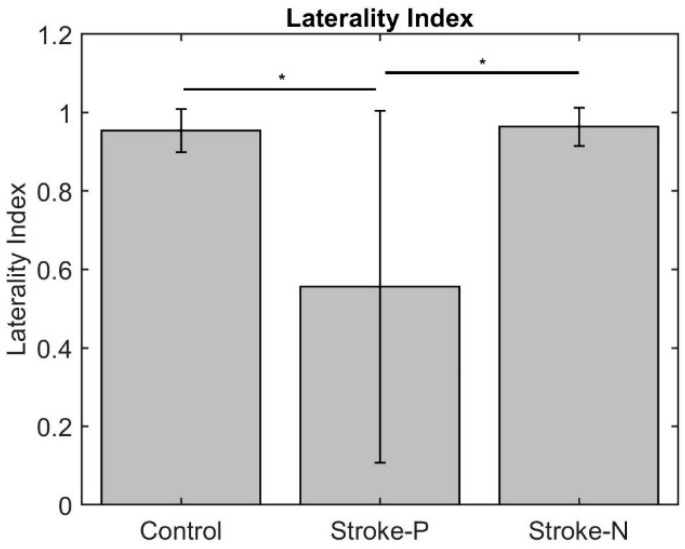
Laterality Index. Stars indicate statistically significant differences among control, stroke paretic hand (Stroke-P), and stroke non-paretic hand (Stroke-N): * < 0.05.

**Figure 8 jcm-11-06449-f008:**
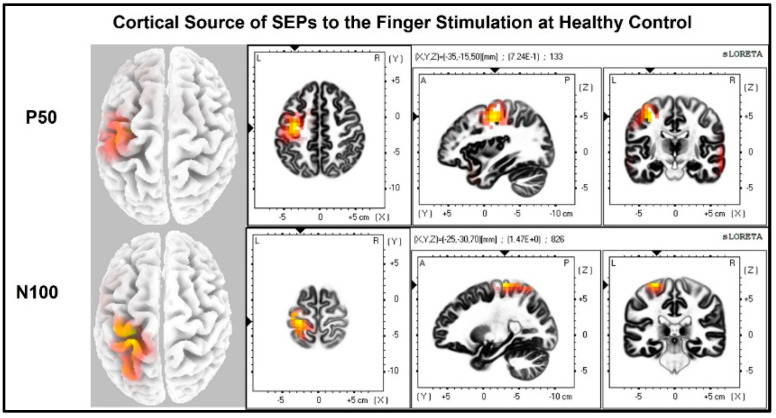
Cortical sources of Somatosensory Evoked Potential (SEP) components in Healthy Control. The right hand was stimulated, and only contralateral (**left**) cortical sources were detected for P50 and N100.

**Figure 9 jcm-11-06449-f009:**
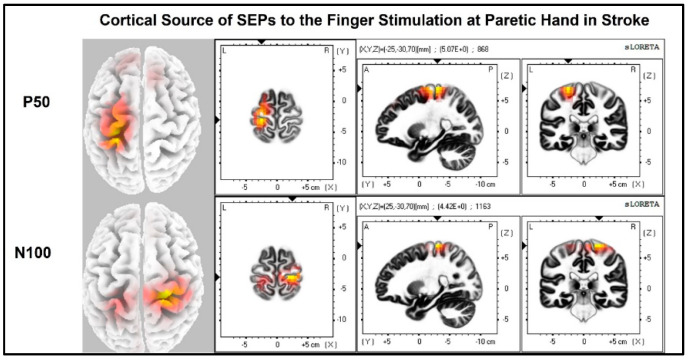
Cortical sources of SEP components in stroke when the paretic hand is stimulated. The paretic (**right**) hand was stimulated, contralateral (**left**) source activities were detected at the time point of P50, and bilateral source activities (more activities in the ipsilateral (**right**) hemisphere) are detected at the time point of N100.

**Figure 10 jcm-11-06449-f010:**
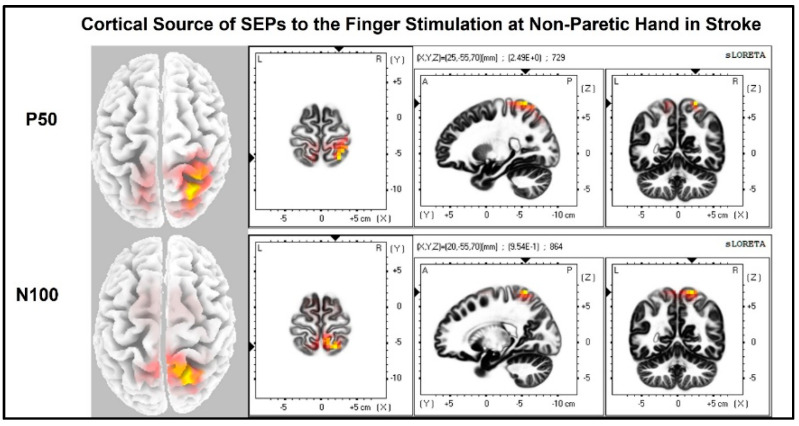
Cortical sources of SEP components in stroke when the non-paretic hand is stimulated. Non-paretic (**left**) hand was stimulated, and contralateral (**right**) source activities were detected mainly at the time points of P50 and N100.

**Table 1 jcm-11-06449-t001:** Stroke participants’ demographics. FM-UE: Fügl–Meyer upper extremity scores.

Subject ID	Lesion Side	Paretic Hand	FM-UE (Total: 66)	Stroke Year
S001	Right	Left	6	2017
S002	Right	Left	63	2019
S003	Left	Right	11	2014
S004	Left	Right	26	2019
S005	Left	Right	63	2013
S006	Left	Right	32	2021
S007	Right	Left	40	2019
S008	Right	Left	19	2021
S009	Left	Right	62	2007

**Table 2 jcm-11-06449-t002:** Descriptive Statistics.

Measure	Mean	Mean	95% CL Lower	95% CL Higher	Std	Min	Max	Median
Latency								
Latency—P50	Stroke-P	71.30	60.12	82.48	13.37	55.00	93.60	70.10
Latency—P50	Stroke-N	57.51	52.46	62.57	6.58	51.40	67.00	54.00
Latency—P50	Control	49.30	45.89	52.71	4.08	42.00	53.80	51.10
Latency—N100	Stroke-P	134.30	112.63	155.97	25.92	87.00	158.00	149.20
Latency—N100	Stroke-N	99.40	86.11	112.69	17.29	78.00	134.40	96.00
Latency—N100	Control	91.13	76.38	105.87	17.64	72.00	119.00	85.80
Amplitude (Amp)								
Amp—P50	Stroke-P	0.49	0.23	0.74	0.33	0.00	0.88	0.62
Amp—P50	Stroke-N	0.75	0.49	1.00	0.33	0.31	1.25	0.69
Amp—P50	Control	0.75	0.39	1.12	0.44	0.13	1.66	0.69
Amp—N100	Stroke-P	−0.69	−1.26	−0.12	0.74	−2.28	0.00	−0.39
Amp—N100	Stroke-N	−0.37	−0.56	−0.17	0.25	−0.85	−0.01	−0.33
Amp—N100	Control	−0.43	−0.84	−0.02	0.49	−1.28	−0.00	−0.20
Laterality Index (LI)								
Mean LI	Stroke-P	0.56	0.15	0.96	0.48	−0.25	1.00	0.68
Mean LI	Stroke-N	0.93	0.81	1.04	0.15	0.54	1.00	0.98
Mean LI	Control	0.95	0.91	1.00	0.06	0.86	1.00	0.99

## Data Availability

Not applicable.

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
