# Peer review of "Cortical Reorganization of Early Somatosensory Processing in Hemiparetic Stroke"

_jcm, 2022, doi:10.3390/jcm11216449_

Round 1
Reviewer 1 Report
Summary
The authors of this study are seeking to build on previous work in understanding how sensory signaling is altered post-stroke and its relationship to motor impairment. The authors measured afferent cortical responses using EEG, in response to peripheral stimulation to the index finger in individuals post-stroke and age-matched healthy controls. The authors found SEP latencies (P50 and N100) to be delayed and have ipsi- and contra-lesioned EEG responses to paretic stimulation. Additionally, the P50 component had a negative correlation with upper extremity Fugl-Meyer scores.
The study adds insight into “how” post-stroke somatosensory reorganization might occur (i.e., the lateralization of SEP’s) vs “if” (i.e., binary responses of SEP’s). Most importantly they demonstrate a relationship between post-stroke cortical somatosensory reorganization and an often used measure of upper extremity function.
Major Issues
- The use paired t-tests instead of utilizing post-hoc testing to control for the family-wise error rate. This may have strictly been nomenclature, but if truly t-tests, there would be an unnecessary hit to power.
- Lines 147-152 and Figures 4 and 5: Please further explain these data and figures. Differentiate between main effects and post hoc testing.
- Figure 4: Does the significance line that crosses all 3 conditions (Control, Stroke-P, and Stork-N) a main effect? Are they different from the lines from Control to Stroke-P (there is no mention of this specific comparison within the text), and Stoke-P to Stoke-N?
- Figure 5: Are the significance bars reflective of the t-test results? Was there no overall main effect?
- The explanation of why post-stroke somatosensory reorganization has occurred is limited to the level of the cortex. Please additionally postulate as to the possibility of subcortical structures and mechanisms contributing to the observed reorganization (e.g., afferent fibers, proprioception, spinal cord, etc.)
Minor Issues
· While it is understood that the P50 and N100 components are reflective of somatosensory responses with little integration from other cortical areas, briefly explain why the authors decided on these specific components.
· Figures 2 and 3: Expound on how these figures add to the reported results. Were these figures only included as visual aids to demonstrate differences in stroke SEP and control components?
o Figure 3: Lines 138-140 specify that ipsilateral responses are shown when the paretic hand is stimulated, but there only appears to be a response when the non-paretic hand is stimulated.
· Figure 8: Specify if the relationship between FMUE and P50 amplitude is of the P50 amplitude in the ipsi-lesioned or contra-lesioned hemisphere.
Reviewer 2 Report
The paper by Willimason et al deals with use of SSEP as a possible biomarker in the spontaneous neural recovery of sensory cortices after a stroke. The paper is interesting an aims at a subject which is not as explored as it should, by using a widespread and relatively available technique.
I have some doubt I would ask to be solved before I can consider the paper suitable for publication.
Authors never stated why they have chosen mid-latency sensory evoked potentials in their work, instead of N20 which is the best-known SEP wave or the long-latency waves. This decision should be better clarified.
The choice of conduct bilateral stimulations in patients and unilateral stimulation on controls remains difficult to understand, if considered together with the fact the preferred-side was not assessed. I believe that this could affect the results on laterality that in fact are difficult to interpreted since it results different in Stroke-P when compared with Stroke-N while one might expect that stimulation on the unaffected side of stroke patients should return a normal SEP signal.
I cannot understand well which tested have been Bonferroni corrected and which have not.
Since authors correctly speculate on the beneficial role as biomarker of this technique suggesting that tdcs or neurostimulation could be used, I would also suggest authors to mention fMV stimulation, which is probably the best way to interact with such modality since it provides directly a sensory input. Authors can find useful literature on this point in Toscano M. Front Neurol. 2019 Feb 19;10:115. doi: 10.3389/fneur.2019.00115
Reviewer 3 Report
-when manuscript is printed there is “and” at the end of the authors' list
- Why N20 was not first determined and localized in order to first determine the exact localization of the somatosensory cortex?
-Can authors discuss the stimulation parameters used in the present study and discuss P50 and N100 relating to previous studies using the same frequency/magnitude of stimulation versus results when using different frequencies/magnitudes. This can also be one of the limitations of the present study that needs to be clarified in the Discussion section.
For example: DOI: 10.1142/S0219519420400151, but please cite more references
-Abstract – raw 25 – it is stated that P50 and N100 latency is delayed in stroke patients when stimulating the paretic hand. Can authors be specific if this delay is related to the contralateral or ipsilateral hemisphere related to stimulation of the paretic hand?
-raw 58-60, please rewrite this sentence and correct the reference /style
-Information on patient therapy is missing if they are having it, some details of duration after stroke or info when the stroke occurred. The second sentence in the paragraph 2. Material and Methods (raw 77-79) is not belonging to the participants data and should be presented in the subsection “Data analysis”, like other parts of the manuscript related to the data analysis.
-Can authors present the intensity level of vibrotactile stimuli for patients vs healthy controls. Are there significant differences between patients vs control healthy subjects?
-It is suggested to include two subsections in the Result section for presenting results related to contralateral data (to stimulated hand) and ipsilateral data (to stimulated hand) since it is hard for a reader to follow the presentation logic of the results in the Results section. If the results are mainly P50 and N100 related to the contralateral response hemisphere (with paretic hand stimulation) then this need to be clearly stated.
- Fig 4 and Fig 5 are these results related to contralateral to stimulated hand P50 and N100 latency ? If yes can this information be updated in Fig 4 and Fig 5 title.
-Fig 6 and Fig 7 are these results related to contralateral to stimulated hand P50 and N100 amplitude? If yes can this information be updated in the Fig 4 and Fig 5 title. If there are no significant differences, it is suggested not to present the data as figures but to state these results in the manuscript results section. Therefore it is suggested to remove Fig 6 and Fig 7.
-raw 163 “Even though the amplitude differences of P50 and N100 were not statistically signif- 162
icance between stroke and control groups (Figures 6-7, P > 0.05), there was a reduced am- 163 plitude of P50 in stroke group when the paretic hand was stimulated”. Where is the statistical proof for this statement/result, figure? Is this reduced amplitude of P50 in stroke group related to contralateral or ipsilateral response when the paretic hand was stimulated? This explanation is given in Discussion but if relevant according to the discussion of authors than it is wise to present it but not present on figures something that is not significant (like fig 6 and 7)?
- “The Pearson corre- 164 lation analysis showed that there was a significant negative linear relationship between 165 P50 amplitude and Fügl-Meyer Upper Extremity (FM-UE) Score (R=-0.630, P=0.047), as 166 shown in Figure 8”. Is this result related to contralateral or ipsilateral amplitude of P50 response when the paretic hand was stimulated? Also, update the clarification in the figure 8 title.
- “Figure 9. Laterality Index. Stars indicate statistically significant differences among control, stroke 186
paretic hand (Stroke-P) and stroke non-paretic hand (Stroke-N): *<0.05 **<0.01 ***<0.001. 187”
If no significant difference was found at **<0.01 ***<0.001, why is this information relevant here?
-“Figure 11. Cortical sources of SEP components in stroke when the paretic side is stimulated.” More precisely, would be non-paretic hand?
- “Figure 11. Cortical sources of SEP components in stroke when the paretic side is stimulated. Pa- 192 retic (right) hand was stimulated, and bilateral source activities were detected with contralateral (ipsile- 193 sional) source at the time point P50 and ipsilateral (contralesional) source at the time point of N100. 194”
It is not quite clear the terminology contralateral (ipsilesional) source and ipsilateral (contralesional) source? If stated in the Fig 11 title that the paretic hand was stimulated, then simple terminology can be used: contralateral hemispheric source response and ipsilateral hemispheric source response. The ipsilateral hemispheric source response is not quite persuasive for P50 in the presented figure 11. Is it this slight frontal activity?
-“Figure 12. Cortical sources of SEP components in stroke when the non-paretic side is stimulated.” More precisely, would be a non-paretic hand?
-Are the cortical responses somehow slightly shifted posteriorly for P50 and N100 when on paretic hand is stimulated?
-raw 200-202 please rewrite this sentence to be more understandable. This statement is somehow weird “while 201 controls had only contralateral cortical responses”
-raw 202-204 “This response was mostly seen in N100 202 when stimulating the paretic index finger. There is also some bilateral activity when stim- 203 ulating the non-paretic index finger.” Please rewrite these sentences to be more understandable.
-raw 208-209, please rewrite this sentence to be more understandable and insert the required references.
“However, to the best of our knowledge, these 208
previous studies have not yet been extended to somatosensory regions of the brain. 209”
-Whay acronym M1 is introduced if it is not used later in the manuscript text?
-raw 229-230 “since the results show largely 229 activation over the lesioned hemisphere for P50 component SEP. 230” Please rewrite this part of the sentence to be more understandable.
-Discussion is missing some concrete information (references) on the usage of vibrotactile stimulation in the rehabilitation of stroke, for example, VTS glove, etc
â–ºdoi: 10.1186/s12984-021-00813-7
â–º doi: 10.1186/s12984-021-00871-x
â–ºWearable vibrotactile stimulation for upper extremity rehabilitation in chronic stroke: clinical feasibility trial using the VTS Glove
Caitlyn E. Seim, Steven L. Wolf & Thad E. Starner
Journal of NeuroEngineering and Rehabilitation volume 18, Article number: 14 (2021)
â–º Brain Sci. 2022, 12, 358. https://doi.org/10.3390/brainsci12030358
-Authors can discuss also the fact of N20 measurement, if this was done in the present study to determine the localization of S1, maybe analysis of P50 and N100 would be somehow easier to interpret.
Round 2
Reviewer 2 Report
Authors have provided interesting replies to my comment, and I want to thank them for their effort. Unfortunately, I still have some doubt.
About the first point, authors explain very well their point. I thank them
In the added parts, I found this sentence. “Stroke participants had a significantly higher sensation threshold than healthy subjects in their paretic hand (two-sample t-test p = 0.025), resulting in higher actual stimulation intensity. There is no significant difference between stroke–paretic hand and healthy controls on the sensation threshold or actual stimulation intensity.” Is that correct or paretic hand in the second sentence is used istead of non-paretic hands? Or is it between hands in stroke patients?
I wonder why authors used a test as GEE because data were not normally dsitruitd and then used pearson correlation analysis instead of non parametric test
In the discussion, authors added possible combination with vibrotactile glove which is still relatively experimental without citing focal muscle vibration which is instead used more and recommended in some guidelines and has a higher level of evidence. In support to this, authors can cite the following papers:
Toscano M Frontiers Neurology 2020 11:567833
Marconi B Neurorehabiliation and Neural repair 2011 25 48
Author Response
Please see our responses as attached. Thank you so much.

Reviewer 3 Report
The authors clarified the raised concerns. Thank you.
Author Response
Thank you so much! We are glad that we have successfully address all the comments. We appreciate your effort on helping us improving this work.